

# Identification of *Salmonella* for public health surveillance using whole genome sequencing

Philip M. Ashton[1], Satheesh Nair[1], Tansy M. Peters[1], Janet A. Bale[1], David G. Powell[1], Anaïs Painset[1], Rediat Tewolde[2], Ulf Schaefer[2], Claire Jenkins[1], Timothy J. Dallman[1], Elizabeth M. de Pinna[1], Kathie A. Grant[1,3] and Salmonella Whole Genome Sequencing Implementation Group

[1] Gastrointestinal Bacterial Reference Unit, Public Health England, London, United Kingdom
[2] Applied Laboratory and Bio-Informatics Unit, Public Health England, London, United Kingdom
[3] Gastrointestinal Infections, NIHR Health Protection Research Unit in Gastrointestinal Infections, London, United Kingdom

Corresponding authors
Claire Jenkins,
claire.jenkins@phe.gov.uk
Timothy J. Dallman,
tim.dallman@phe.gov.uk

## ABSTRACT

In April 2015, Public Health England implemented whole genome sequencing (WGS) as a routine typing tool for public health surveillance of *Salmonella*, adopting a multilocus sequence typing (MLST) approach as a replacement for traditional serotyping. The WGS derived sequence type (ST) was compared to the phenotypic serotype for 6,887 isolates of *S. enterica* subspecies I, and of these, 6,616 (96%) were concordant. Of the 4% ($n = 271$) of isolates of subspecies I exhibiting a mismatch, 119 were due to a process error in the laboratory, 26 were likely caused by the serotype designation in the MLST database being incorrect and 126 occurred when two different serovars belonged to the same ST. The population structure of *S. enterica* subspecies II–IV differs markedly from that of subspecies I and, based on current data, defining the serovar from the clonal complex may be less appropriate for the classification of this group. Novel sequence types that were not present in the MLST database were identified in 8.6% of the total number of samples tested (including *S. enterica* subspecies I–IV and *S. bongori*) and these 654 isolates belonged to 326 novel STs. For *S. enterica* subspecies I, WGS MLST derived serotyping is a high throughput, accurate, robust, reliable typing method, well suited to routine public health surveillance. The combined output of ST and serovar supports the maintenance of traditional serovar nomenclature while providing additional insight on the true phylogenetic relationship between isolates.

## INTRODUCTION

The *Salmonella*e are major human pathogens and represent a significant global public health issue causing morbidity and mortality resulting in a high social and economic burden worldwide (*Majowicz et al., 2010*). The genus consists of two species; *Salmonella enterica* and *S. bongori*. There are six subspecies of *S. enterica* differentiated by biochemical

variations, namely subspecies *enterica* (I), *salamae* (II), *arizonae* (IIIa), *diarizonae* (IIIb), *houtenae* (IV) and *indica* (VI) (*Threlfall, Ward & Old, 1999*). Subspecies I, *S. enterica* subsp. *enterica* cause 99% of human and animal infections. The two main pathologies associated with *S. enterica* are gastroenteritis and typhoidal disease. The typhoidal *Salmonella*e include *S.* Typhi and *S.* Paratyphi A, B and C. They are host restricted, monophyletic, rarely undergo recombination events and exhibit convergent evolution driven by genome degradation (*Wain et al., 2015*). The majority of gastroenteritis in the UK is caused by the host generalist serovars, such as *S.* Typhimurium and *S.* Enteritidis, and host adapted serovars that are adapted to a specific animal reservoir but can infect man and include *S.* Dublin, *S.* Gallinarum *S.* Choleraesuis, and *S.* Bovismorbificans (*Langridge et al., 2015*).

Approximately 8,000 isolates are referred to the *Salmonella* Reference Service (SRS) at Public Health England (PHE) each year from local and regional hospital laboratories. In April 2015, PHE implemented whole genome sequencing (WGS) as the routine typing tool for public health surveillance of *Salmonella* infections. Prior to April 2015, presumptive *Salmonella* isolates referred to SRS were speciated and sub-speciated using PCR (*Hopkins et al., 2009*; *Hopkins et al., 2011*) and grouped into serovars as described in the White-Kauffman-Le Minor scheme (*Grimont & Weill, 2007*; *Guibourdenche et al., 2010*; *Issenhuth-Jeanjean et al., 2014*). This methodology is based on reactions of rabbit antisera to the lipopolysaccharide (O antigen encoded by *rfb* genes) and flagellar antigens (phases 1 and 2 of H antigen encoded by *fli*C and *flj*B). The scheme utilises this phenotypic variation, expressed as an antigenic formulae, to divide *Salmonella* into more than 2,600 serovars. Epidemiological investigations of *Salmonella* infecting humans and animals have relied on serotyping for over 70 years; national and international governmental agencies base guidelines and regulations on the serotyping method and the use of this nomenclature is a globally recognised form of communication (*Swaminathan et al., 2006*; *EFSA, 2010*). Furthermore, serovars have often been shown to correlate with host range and disease sequelae (*Gordon, 2011*; *Wain et al., 2015*; *Langridge et al., 2015*).

There are, however, a number of issues with the serotyping approach; specifically, the expense and expertise required to produce the antisera and, furthermore, serotyping does not reflect the genetic relatedness between serovars, nor does it provide an evolutionary perspective. Alternative molecular serotyping methods have been described previously including Pulsed-field gel electrophoresis, ribotyping, repetitive extragenic palindromic sequence-based PCR (rep-PCR) and combined PCR- and sequencing-based approach that directly targets O- and H-antigen-encoding genes (*Ranieri et al., 2013*; *Shi et al., 2015*). In 2012, *Achtman et al. (2012)* proposed a sequenced based approach, multilocus sequence typing (MLST), based on the sequences of multiple house-keeping genes. Isolates that possess identical alleles for the seven gene fragments analysed are assigned a common sequence type (ST) and related STs from clonal complexes are termed e-Burst Groups (eBGs). They showed that ST and eBGs strongly correlated with serovar and so utilising this approach would facilitate backward compatibility with historical data, minimise disruption for reference laboratory service users and facilitate data exchange with other colleagues in the field.

Advances in whole genome sequencing (WGS) methodologies have resulted in the ability to perform high throughput sequencing of bacterial genomes at low cost making WGS an economically viable alternative to traditional typing methods for public health surveillance and outbreak detection (*Koser et al., 2012*). Whilst WGS provides the opportunity to resolve bacterial strains to the single nucleotide resolution needed for identifying cases linked to a common source of infection (*Dallman et al., 2015*), grouping isolates into higher taxonomical clones (e.g., those defined by serotyping) is an important step. The decision to adopt WGS as a routine typing method at PHE provided the opportunity to review our approach to typing *Salmonella* and to implement the MLST approach in parallel with WGS.

The aim of this study was to evaluate MLST, as derived from WGS data, as a replacement for conventional serotyping of *Salmonella* for routine public health surveillance and to provide insight into the genetic population structure of all *Salmonella* species in England and Wales during a 12 month period.

## METHODS

### Bacterial strains

All isolates ($n = 7,465$) of *Salmonella* from human cases of gastrointestinal disease submitted to SRS from local and regional hospital laboratories in England & Wales between 1st April 2014 and 31st March 2015 were sequenced in parallel with phenotypic serotyping (Table S1). Of these, 7,338 were identified as subspecies I and included 263 different serovars. The ten most common serovars in this dataset were Enteriditis (2,310), Typhimurium (1,407), Infantis (184), Typhi (184), Newport (173), Virchow (162), Kentucky (160), Stanley (146), Paratyphi A (135) and Java (99). One hundred and twenty seven isolates were identified as subspecies II–IV (*S. enterica* subspecies *salamae* $n = 28$; *S. enterica* subspecies *arizonae* $n = 25$; *S. enterica* subspecies *diarizonae* $n = 49$; *S. enterica* subspecies *houtenae* $n = 20$) and there was one isolate of *S. bongori*. No isolates belonging to subspecies VI (*S. enterica* subspecies *indica*) were submitted to SRS during the study period.

### DNA extraction for WGS

DNA extraction of *Salmonella* isolates was carried out using a modified protocol of the Qiasymphony DSP DNA midi kit (Qiagen). In brief, 0.7 ml of overnight *Salmonella* culture in a 96 deep well plate was harvested. Bacterial cells were pre-lysed in 220 µl of ATL buffer (Qiagen) and 20 µl Proteinase K (Qiagen), and incubated shaking for 30 min at 56 °C. Four µl of RNase at 100 mg/ml (Qiagen) was added to the lysed cells and re-incubated for a further 15 min at 37 °C. This step increases the purity of the DNA for further downstream sequencing. Extraction of DNA from the treated cells was performed on the Qiasymphony SP platform (Qiagen) and eluted in 100 µl of water. DNA concentration using the GloMax system (Promega) was determined for the following sequencing steps.

### DNA sequencing

Extracted DNA was then processed using the NexteraXT sample preparation method and sequenced with a standard 2x101 base protocol on a HiSeq 2500 Instrument in fast mode (Illumina, San Diego, CA, USA).

## Bioinformatics workflow

FASTQ reads were quality trimmed using Trimomatic (*Bolger, Lohse & Usadel, 2015*) with bases removed from the trailing end that fell below a PHRED score of 30. If the read length post trimming was less than 50 bp the read and its pair were discarded. The PHE KmerID pipeline (https://github.com/phe-bioinformatics/kmerid) was used to compare the sequenced reads with 1,769 published genomes to identify the bacterial species (and *Salmonella* subspecies) and to detect cultures submitted by the local and regional hospital laboratories that contained more than one bacterial species (mixed cultures). KmerID determines a similarity index between the FASTQ reads and each of the 1,769 published reference genomes by calculating the percentage of 18-mers in the reference that are also present in the FASTQs. Only 18-mers that occur at least twice in the FASTQ are considered present. Mixed cultures are detected by comparing the list of similarities between the sample and the references with the similarities of the references to each other, and filtering this comparison for inconsistencies. ST assignment was performed using the Metric Orientated Sequence Typer (MOST), a modified version of SRST (*Inouye et al., 2012*), available from https://github.com/phe-bioinformatics/MOST. The primary difference between SRST and MOST is in the metrics provided around the result, while SRST gives a single score, MOST provides a larger array of metrics to give users more details on the read level associated with their result. Preliminary analysis was undertaken using the MLST database described in *Achtman et al. (2012)*. It takes approximately 10–15 min to run MOST using a single core on the PHE infrastructure which consists of Intel Xeon CPU E5-2680 0@ 2.70GHz, 16 cores sharing 125 Gb Memory.

For isolates that had novel STs, or a ST but no associated serovar in the Achtman MLST database, the serovar was determined by phenotypic serotyping at PHE. STs and corresponding serovars of isolates serotyped and sequenced during this study were added to a modified version of the Achtman MLST database, held and curated at PHE. These novel STs were assigned a preliminary ST (PST) and an inferred serovar was determined. The PHE MLST database currently holds 7,000 strains and 1,200 serovars and is up-dated every three months.

For some STs that contained two serotypes, whole genome SNP phylogenetic analysis was carried out by mapping the strains of interest against a reference genome from within the same sequence type (for ST909 H145100685 was used; for ST49, H143720759 was used), using BWA mem (*Li & Durbin, 2009*). SNPs were called using GATK2 (*DePristo, Banks & Poplin, 2011*) in unified genotyper mode. Core genome positions that had a high quality SNP (>90% consensus, minimum depth $10\times$, $GQ \geq 30$, $MQ \geq 30$) in at least one strain were extracted and RAxML v8.1.17 phylogenies determined with the gamma model of rate heterogeneity and 100 bootstraps undertaken.

## RESULTS

*Achtman et al. (2012)* described the population structure of *Salmonella enterica* as monophyletic lineages of STs that have evolved from a single founder node and termed these discrete clusters eBGs. The population structure of all the *Salmonella* species submitted

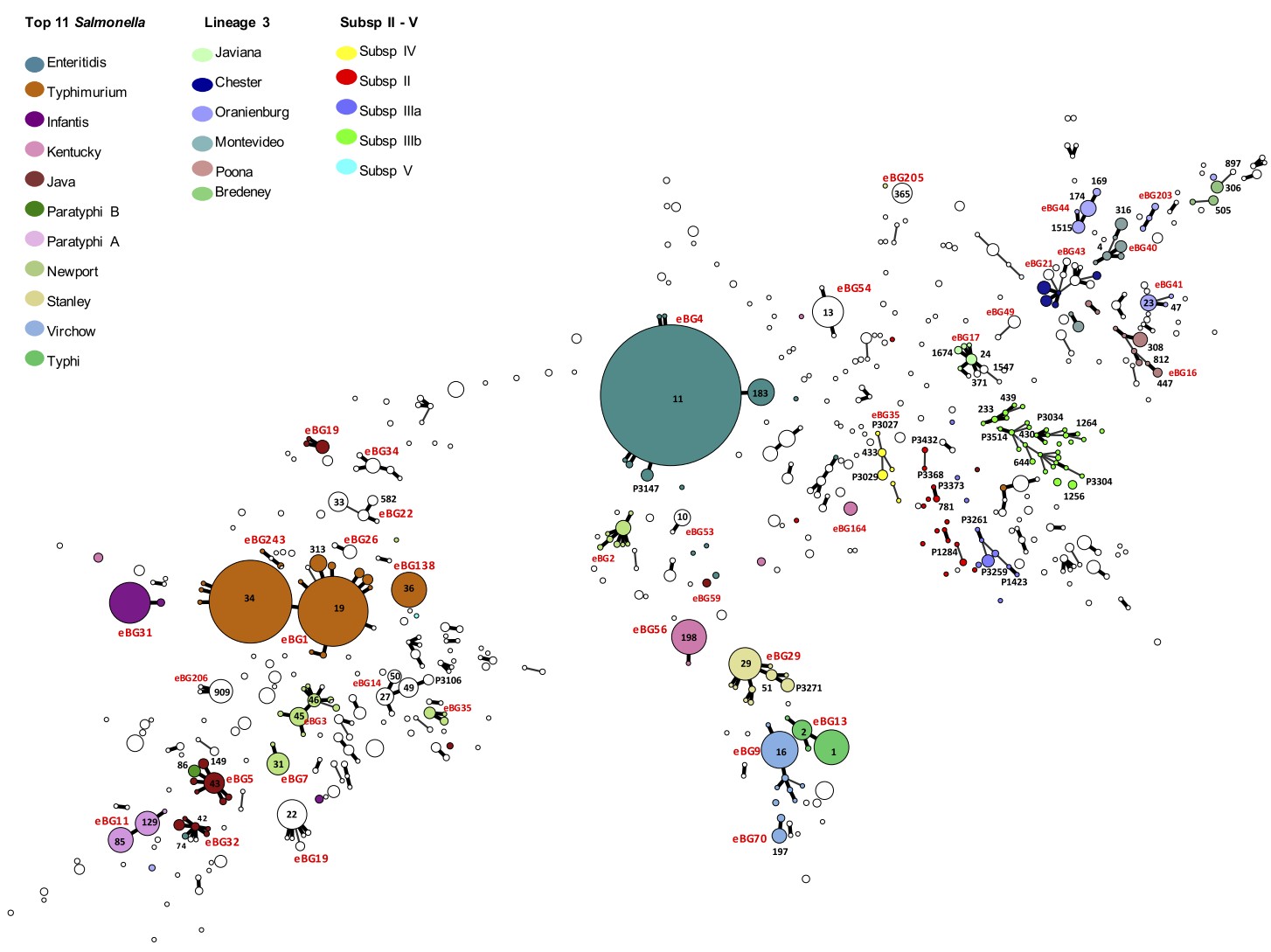

**Figure 1** Population structure of all *Salmonella enterica* isolates submitted to PHE from local and regional hospital laboratories in England and Wales between April 2014 and March 2015 (see Table S1 for details).

to PHE between April 2014 and March 2015 is illustrated by the minimum spanning tree in Fig. 1.

### *Salmonella* subspecies I

The ST and corresponding serovar designation obtained from the MLST database were used to compare the WGS derived ST to the phenotypic serotype for 6,887 (94%) of 7,338 isolates of subspecies 1, and of these, 6,616 (96%) had the same result by both methods (Table S1). It was not possible to compare phenotypic serotyping with MLST-based serotyping for 451 (6%) subspecies I isolates because either the phenotypic serotype could not be determined due to an incomplete antigenic structure (*S.* Unnamed) (*n* = 423) or the serovar could not be determined because the ST did not have a designated serotype in the MLST database (*n* = 70). Forty-two isolates were both *S.* Unnamed and had no MLST designated serotype.

For the 423 (5.8%) subspecies I isolates reported as *S.* Unnamed, 318 (90%) were designated a serotype from the WGS derived MLST data. The most common serovars typed in this way included *S.* Typhimurium (118), *S.* Virchow (30), S. Stanley (17), *S.* Enteritidis (16), *S.* Infantis (14) and *S.* Thompson (13). Of the 7,338 strains tested, 70 (1%) had no serotype designation in the MLST database, of which 28 (40%) were serotyped phenotypically (Table S1).

### Subspecies I novel sequence types

Novel sequence types that were not present in the MLST database were identified in 8.6% ($n = 654$) of the strains (Table S1). These 654 isolates belonged to a total of 326 novel STs, designated PST; the modal number of isolates identified per PST was one (Fig. 2A). There was no difference in the distribution of number of isolates per PST depending on whether the PST had a known serovar or belonged to an unnamed or ambiguous serotype. The rate at which PSTs were received throughout the year was plotted and revealed a linear relationship ($R^2 = 0.98$, $y = 1.04 * x$, where $x = $ number of days since April 1st 2014) (Fig. 2B).

The serovars with the highest number of new PSTs were *S.* Typhimurium ($n = 9$), *S.* Stanley ($n = 9$), *S.* Enteritidis ($n = 9$) and *S.* Newport ($n = 8$), although the majority of these PSTs were single locus variants (SLVs) of established STs, belonging to these serovars (*S.* Typhimurium 8/9, *S.* Stanley 7/9 and *S.* Newport 7/8). There were also serovars for which a large number of PSTs were identified that were not SLVs of established STs (*S.* Agama 5/5, *S.* Agbeni 5/5, *S.* Saint-Paul 5/5, *S.* Enteritidis 4/9) which may represent new eBGs that share these serotypes.

### Subspecies I mismatches

Four percent ($n = 271$) of the isolates tested exhibited a mismatch between the WGS MLST derived serovar and the phenotypic serotyping results. Of the 271 mismatches, 119 were due to a process error in the laboratory either in the phenotypic serotyping or the DNA extraction part of the WGS pipeline. With respect to the phenotypic serotyping, common errors included mislabelling samples and misinterpreting or incorrectly transcribing the antigenic structure, especially when the antigenic structures were similar. For example, *S.* Agona (I 4, 12:f, g, s:-) and *S.* Derby (I 4, 12:f, g:-). DNA extraction errors were associated with mislabelled samples.

Twenty-six mismatches were potentially caused by the predicted serotype designation in the Achtman MLST database being incorrect which may be attributed to single entries that had been misidentified at the laboratory from which the MLST data was submitted. For example, in the original database ST1499 is represented by one entry identified by the submitter as *S.* Litchfield. Subsequently, phenotypic serotyping at PHE identified this ST as *S.* Bovis-morbificans in more than five isolates. ST1499 belongs to eBG34 which comprises two other STs both associated with *S.* Bovis-morbificans, indicating that the original entry in the MLST database is likely to be incorrect.

The most common reason for mismatches occurring between the WGS MLST derived serotype and the phenotypic serotype ($n = 126$) occurred when two different serovars belonged to the same eBG and the same ST (see Table 1 and discussed in more detail below).

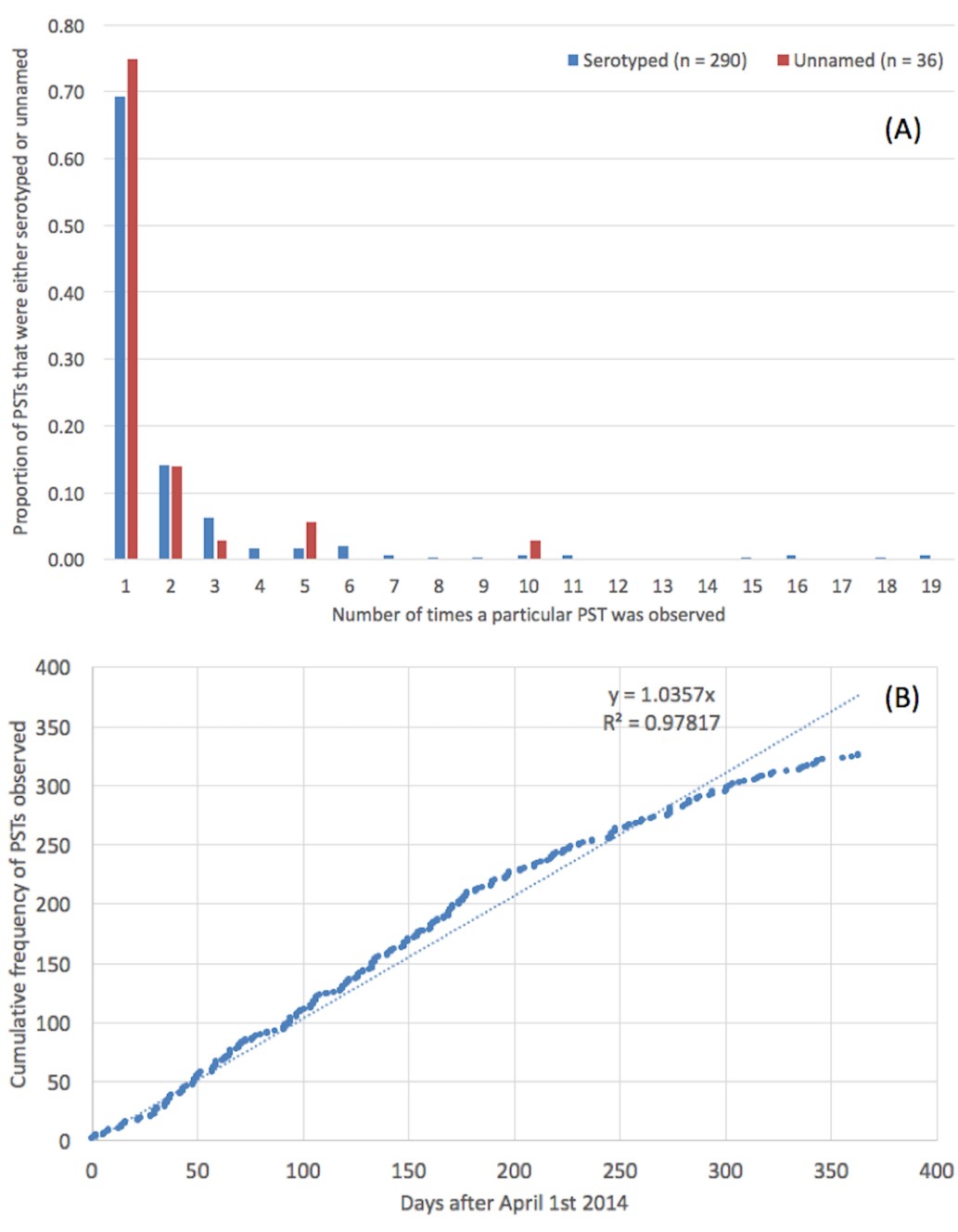

**Figure 2  Trends in preliminary sequence types.** (A) Novel, preliminary STs (PST) and the modal number of isolates identified per PST. (B) The rate at which PSTs were identified throughout the time frame of the study.

Ashton et al. (2016), *PeerJ*, DOI 10.7717/peerj.1752

**Table 1** Examples where two serovars belonged to the same eBG and the same ST.

| Serotype | Antigenic structure | ST |
|---|---|---|
| Bareilly | I 6,7: y: 1,2 | 909 |
| Richmond | I 6,7: y: 1,5 | |
| Saintpaul | I 4,5,12: e,h: 1,2 | 49 |
| Haifa | I 4,5,12: z,10: 1,2 | |
| Sandiego | I 4,12: I,v: e,n,z15 | 20 |
| Brandenburg | I 4,12: e,h: e,n,z15 | |
| Uganda | I 3, 10: I,z13: 1,5 | 684 |
| Sinstorf | I 3, 10: I,v: 1,5 | |
| Agona | I 4,12: f,g,s:- | 13 |
| Essen | I 4,12: f,g,m:- | |
| Napoli | I 1,9,12: I,z13: enx | P3141 |
| Zaiman | I 1,9,12: I,v: enx | |

For 157 of the 271 mismatches, assembly based ST calls were available from Enterobase (http://enterobase.warwick.ac.uk/). We compared the STs called by our mapping pipeline and the assembly based Enterobase pipeline and the same ST was called in 100% of samples.

## Serovars Enteritidis and Dublin

Of the 2,308 isolates of *S*. Enteritidis identified by both phenotypic serotyping and WGS MLST derived serotyping, 2,296 belonged to eBG4, including 2,200 ST11 and 76 ST183 (Fig. 1). There were five additional SLVs of ST11, four of which were novel types. *S*. Gallinarum and *S*. Pulloram can be difficult to distinguish from *S*. Enteritidis (*Thomson et al., 2008*) but neither of these serovars were identified in this study. Serologically, *S*. Dublin ([1],9,12:g,p:-) has a similar antigenic structure to *S*. Enteritidis ([1],9,12:g,m:-), and in *Achtman et al. (2012)*, eBG32 (ST74) contained both *S*. Enteritidis and *S*. Dublin. However, in this study both isolates belonging to ST74 eBG32 typed as *S*. Enteritidis. Of the 2,308 isolates, 26 belonged to nine new PSTs. The most common was P3147, a previously undescribed SLV of ST11, comprising 16 cases including 10 known to have travelled to Malaysia or Singapore.

## Serovar Typhimurium

In this study, eBG1 contained 1,392 isolates of *S*. Typhimurium and monophasic *S*. Typhimurium (rough and non-motile variants) (*Hopkins, De Pinna & Wain, 2012*). The monophasic variants also belong to eBG138 (primarily ST 36) and eBG243. In contrast to eBG1 described in *Achtman et al. (2012)*, which was represented by a large central ST19 node with at least 27 SLV STs comprising much smaller numbers of strains, eBG1 in the PHE dataset shows a predominance of both ST19 and ST34 and less allelic variation. Only nine SLVs to ST19 were identified including three undesignated STs (Fig. 1).

## Serovars Java/Paratyphi B data

Despite the different disease outcomes associated with *S*. Paratyphi B (most commonly associated with invasive disease and paratyphoid fever) and *S*. Java (most commonly

**Table 2  Differences in ST between S. Java and S. Paratyphi B.** *S.* Java isolates in this study belonged to a diverse range of eBGs and STs associated with *S.* Java whereas *S.* Paratyphi B belonged to ST86 only.

| Phenotypic serovar | eBG5 | | | | | eBG 9 | eBG59 | eBG32 | | eBG95 |
| | ST43 | ST149 | ST307 | ST1577 | ST86 | ST88/127 | ST28 | ST423 | ST682/1588 | 1583 |
|---|---|---|---|---|---|---|---|---|---|---|
| Java | 45 | 7 | 4 | 3 | 2 | 18 | 6 | 5 | 2 | 1 |
| Paratyphi B | 0 | 0 | 0 | 0 | 12 | 0 | 0 | 0 | 0 | 0 |

associated with gastroenteritis) it is not possible to differentiate the two serotypes by serotyping alone. *S.* Java and *S.* Paratyphi B are therefore differentiated in the laboratory by their ability to ferment dextrorotatory tartrate (*S.* Java *d*Ta+ and *S.* Paratyphi B *d*Ta−) (*Malorny, Bunge & Helmuth, 2003*).

The 99 isolates identified by both phenotypic serotyping and WGS MLST derived serotyping as *S.* Java, belonged to a diverse range of eBGs, STs and PSTs (Table 2 and Fig. 1). Two of these 99 isolates belonged to ST86 and the predicted serotype from the MLST database was *S.* Paratyphi B. One of these isolates was from a blood culture (associated with invasive disease) and, therefore, likely to have been misidentified phenotypically. All 12 isolates identified as *S.* Paratyphi B phenotypically, were identified as *S.* Paratyphi B ST86 by WGS MLST.

## Subspecies II–IV and *S. bongori*

Isolates from subspecies II, III, IV, VI and *S. bongori* were not well represented in Achtman's MLST database and thus the majority of isolates from these sub-species sequenced in this study did not belong to a previously designated eBG or ST. The population structure of the 127 non-subspecies I isolates differs markedly from that of *Salmonella enterica* (subspecies 1) (Fig. 1) and shows some similarity to the population structure of lineage 3 in being a connected network of STs.

Sixteen of the 28 isolates belonging to subspecies II were previously designated *S.* Unnamed and the 28 strains belonged to 20 different STs. There were 25 isolates classed as subspecies IIIa (belonging to 10 different STs) and 49 in subspecies IIIb (belonging to 27 different STs). Of the 20 isolates identified as subspecies IV, 10 were designated *S.* Wassenaar (P3029) by phenotypic serotyping and the 20 isolates belonged to five different STs. All isolates of subspecies II–IV and *S. bongori* were correctly speciated using the k-mer ID approach.

## Population structure

As highlighted by *Achtman et al. (2012)*, the majority of isolates in the dataset belong to eBGs that have a one-to-one relationship with a specific serovar including *S.* Typhi, *S.* Paratyphi A and *S.* Heidelberg. In this study, of the serovars comprising more than 25 isolates, there were 17 serovar specific eBGs and 10 examples of a single serovar being associated with multiple eBGs (Fig. 1). There were at least six examples of more than one serovar belonging to the same eBG but different STs, for example *S.* Hadar (ST33) and *S.* Kottbus (ST582) both belong to eBG22 and *S.* Bredeney (ST306) and *S.* Schwarzengrund (ST96) both belong to eBG33 (Table S1).

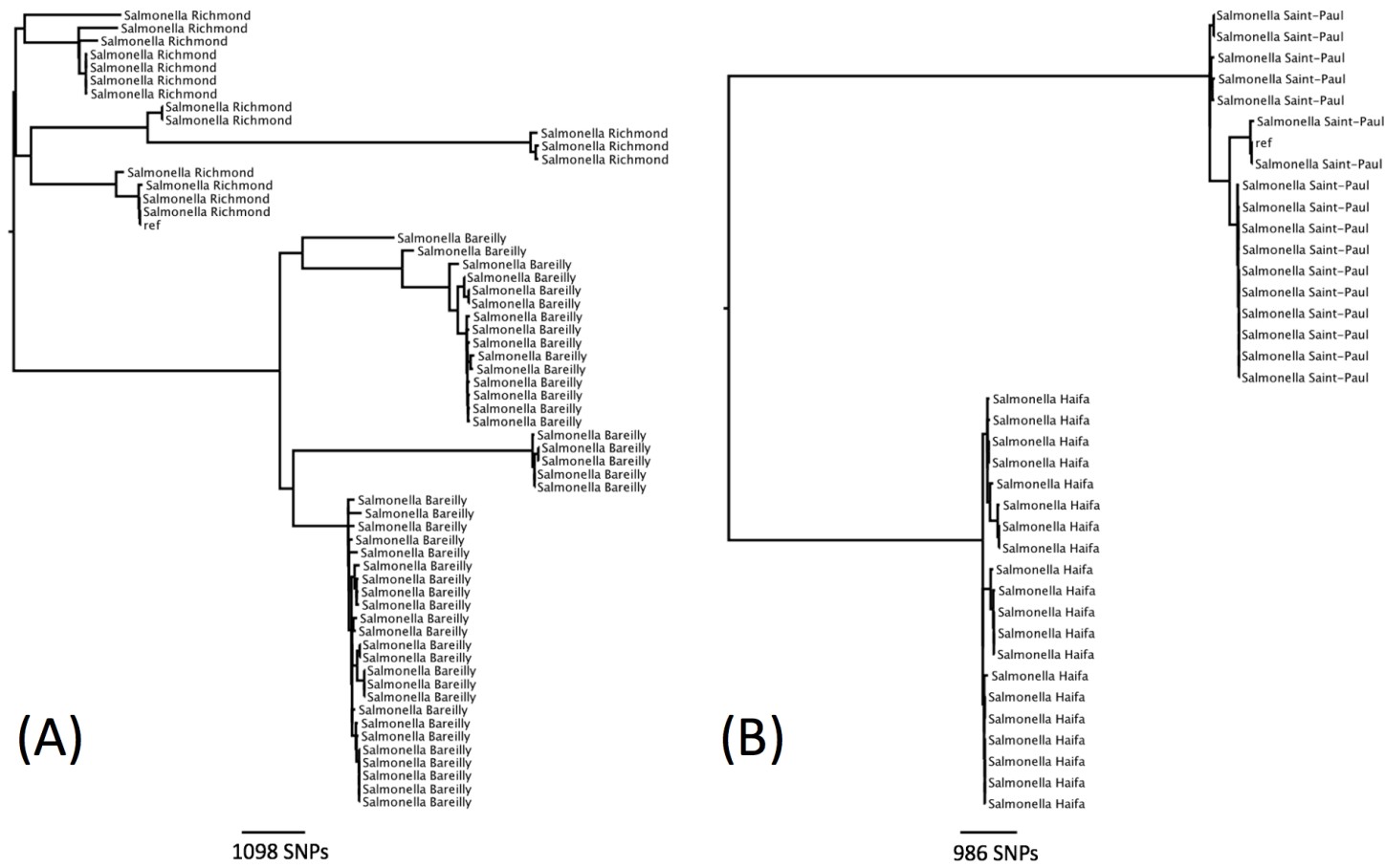

1098 SNPs

986 SNPs

**Figure 3  Phylogenetic relationship within two sequence types containing multiple serotypes.** (A) Phylogenetic relationship of *S*. Richmond and *S*. Bareilly (ST909). (B) Phylogenetic relationship of *S*. Saintpaul and *S*. Haifa (ST49).

There were seven examples where two serovars belonged to the same eBG and the same ST (Table 1). In all of these examples, the antigenic structures of the two serovars were similar with only one antigen differentiating the two serovars. Further analysis was carried out on two examples to determine whether this difference in antigenic structure represented a true difference in strain relatedness or a random change that is not reflected in phylogeny (for example, the insertion of phage encoded antigen). The analysis showed that the change in antigenic structure in *S*. Richmond (I 6,7: y: 1,2) and *S*. Bareilly (I 6, 7: y: 1,5), both ST 909, and in *S*. Saintpaul (I 4,5,12: e,h: 1,2) and *S*. Haifa (I 4,5,12: z,10: 1,2), both ST49, reflected a true phylogenetic difference (Figs. 3A and 3B).

The same higher strata population structure referred to as lineage 3 for *S. enterica* subspecies I, as described by *Achtman et al. (2012)*, was observed in this dataset (Fig. 4). Genomes of these *Salmonella*e are in constant flux and homologous recombination among unrelated eBGs is frequent (*Achtman et al., 2012*; *Didelot et al., 2011*). Serovars in this lineage mainly consists of multiple eBGs and are polyphyletic by nature. *Achtman et al. (2012)* suggested that the population structure of lineage 3 does not comprise of independent startbursts, as observed with other serovars of subspecies I, but rather a

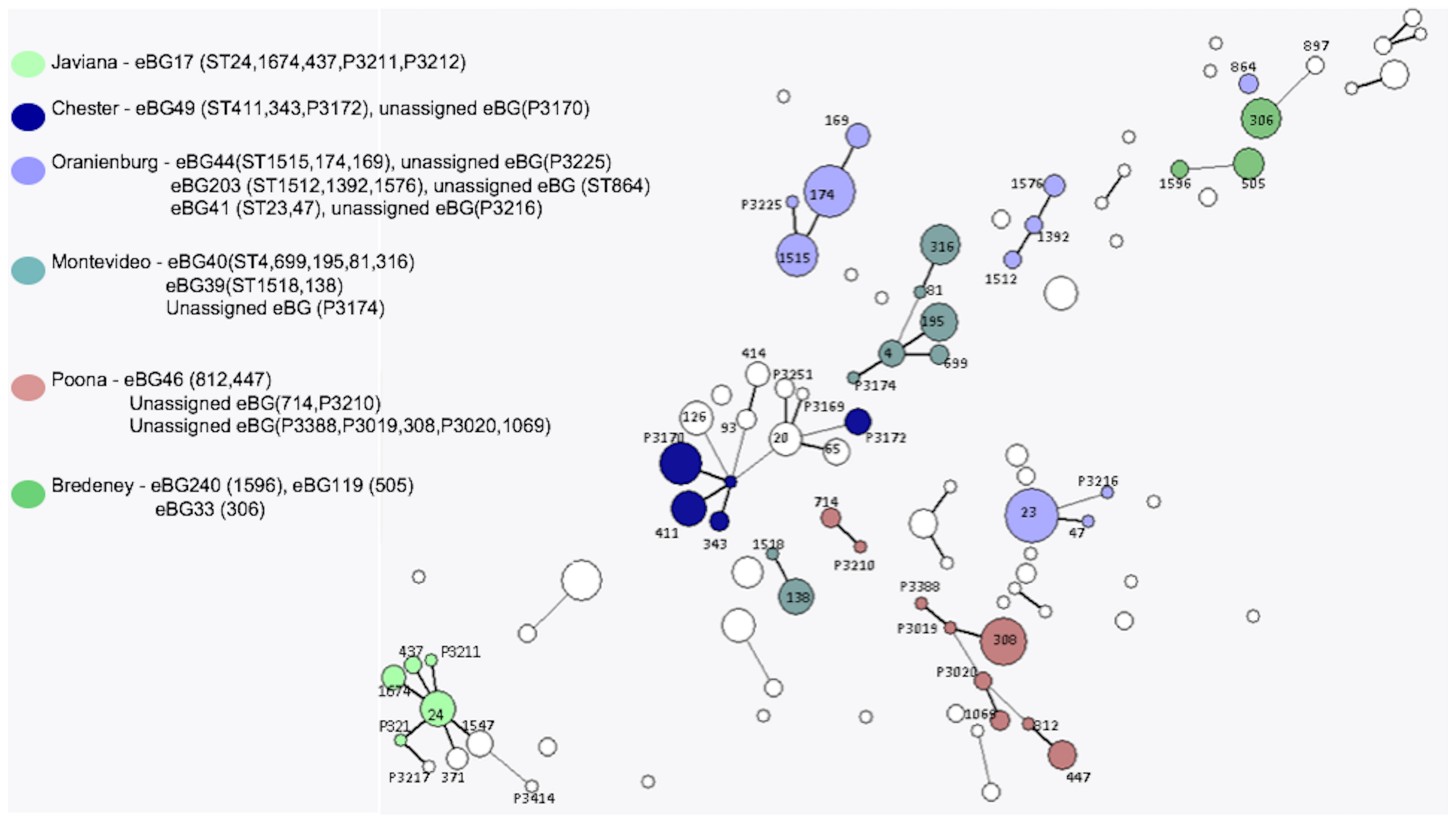

**Figure 4** *Salmonella* **lineage 3 population structure.** Serovars in lineage 3 mainly consist of multiple eBGs and are polyphyletic by nature.

connected network (Fig. 4). The five most common examples of this in the current study, were *S.* Oranienburg, *S.* Montevideo, *S.* Chester, *S.* Poona and *S.* Bredeney (Fig. 4 and Table S1). These five serovars are not represented in the top 10 serovars submitted to SRS during this surveillance period.

## K-mer identification

There were 249 cultures submitted to SRS by the local hospital and regional laboratories for *Salmonella* typing that were a mix of *Salmonella* and non-*Salmonella* species. These were identified by the k-mer identification step and included 138 *Escherichia coli*, 40 *Morganella morganii*, 11 *Citrobacter species* and four *Escherichia albertii*.

## DISCUSSION

In their seminal 2012 paper Achtman and colleagues (*2012*) argued convincingly for replacing serotyping with a MLST approach based on genetic population groupings for typing *S. enterica*. The key aspects of this approach that led PHE to adopt this strategy were (i) the robustness of the population structure as defined by the natural eBG clusters (ii) the fact that eBG designation provides an accurate representation of strain relatedness and (iii) that this approach lends itself to automation. At the same time, it was necessary for PHE to maintain serovar nomenclature in order to facilitate data exchange with other colleagues

in the field and maintain backward compatibility with historical data. It was suggested that by using the MLST approach to infer serovar, and by reporting both inferred serovar and ST, it would be possible to utilise the advantages of both methods and implement a state-of-the-art typing system while keeping disruption for reference laboratory service users to a minimum.

The PHE dataset of 6,887 subspecies I isolates that were serotyped using both traditional phenotypic methods and a derived serotype based on MLST data extracted from the genome during a 12-month time frame, provided further evidence of the robustness of the ST/eBG approach to typing. The 96% concordance between the two techniques in a reference laboratory setting is evidence of the validity and suitability of this approach. There were 451 isolates that had to be excluded from the comparison because both types of data (phenotypic and genotypic) were not available. Of these, for 94% of the isolates, it was the phenotypic serotype that could not be determined indicating that WGS MLST derived serotyping is more robust.

The PHE dataset included single serovars associated with multiple eBG, for example *S.* Typhimurium and *S.* Newport (*Sangal et al., 2010*; *Achtman et al., 2012*) and multiple serovars belonging to the same eBG but with different STs, for example *S.* Java (ST43) and *S.* Paratyphi (ST86) both belong to eBG5 (*Achtman et al., 2012*). In both these scenarios, the correct serovar was determined from the MLST WGS data and the combination of serovar and ST/eBG provided insight into the true phylogenetic relationship between isolates. This data clearly supports Achtman and colleagues argument that eBG and ST designation provides a more accurate representation of strain relatedness than the traditional serovar designation. The phenomenon of multiple serovars belonging to the same ST (for example *S.* Richmond/*S.* Bareilly and *S.* Haifa/*S.* Saintpaul) was a rare but important example of serotyping providing a higher level of strain discrimination within a ST. These strains could be differentiated *in silico* using a tool to infer serovar from the genes that determine antigenic struture, such as seqsero (*Zhang et al., 2015*).

Despite the implementation of WGS, a limited phenotypic serotyping facility continues to be maintained at PHE in order to serotype isolates that cannot be matched to a serovar; either because the ST in the MLST database has no serovar designation or the ST is a novel type. Additionally, it ensures that we maintain the ability to perform the standard reference method for serotyping *Salmonella*. The PHE MLST database is regularly up-dated to include STs recently matched to a serotype by linking the ST to PHE phenotypic serotyping data and novel PSTs. This approach was adopted because at the time of analysis, the Achtman MLST database was not accepting submissions generated by WGS. There was no decrease in the rate at which PSTs were observed during the 12 month study period and the majority of PSTs were only sampled once in that time frame. Many PSTs were SLVs of known STs, indicating that we have not yet sampled the full diversity of known eBGs. New PSTs, not part of any previously identified eBG, were also observed and further diversity was found within *S. enterica* subspecies II–IV and the lineage 3 population. This suggests that there is a large amount of previously unidentified diversity within the species *Salmonellae* associated with both domestically acquired and travel related gastrointestinal disease in human cases resident in England and Wales.

Isolates exhibiting monophasic properties that could not be fully serotyped phenotypically because they had an incomplete antigenic structure were matched to a ST derived serotype. The monophasic variants in this study mainly belonged to eBG1, eBG138 and eBG243 and previous studies have also shown that monophasic variants of *S.* Typhimurium have emerged as a result of multiple independent genetic events (*Soyer et al., 2009*; *Switt et al., 2009*; *Tennant et al., 2010*). Strains with monophasic properties are reportable to European Centre for Disease Prevention and Control (ECDC) but cannot be determined using the ST approach. Alternative strategies for determining monophasic characteristics by PCR are available (*Prendergast et al., 2013*) and methods for extracting this information from the genome sequencing data have been developed at PHE (P Ashton & A Lewis, pers. comm., 2015).

In contrast to *S.* Typhurmurium, where ST could not be used to determine monophasic characteristics, in this study ST was able to differentiate the complex relationship between *S.* Java (Hazard Group (HG) 2 organism) and *S.* Paratyphi B (HG3) with the latter belonging to either ST42 or ST86. If this ST designation proves to be robust, MLST will facilitate the diagnosis of invasive disease and life threatening paratyphoid fever.

The MLST derived serovar correlated well with the traditional serovar designation and demonstrated many advantages over traditional phenotypic serotyping. Monophasic strains with incomplete antigenic structures were accurately assigned to serotypes. Phenotypic serotyping errors, such as misinterpreting or incorrectly transcribing the antigenic structure, were avoided. Novel types were identified, confirmed and given a PST designation. Finally, this approach lends itself to automation and rapid, high-throughput processing.

Two main issues arose during the evaluation of the MLST approach: (i) a number of STs did not have a serovar designation in the MLST database (including subspecies II–IV) and (ii) the unexpectedly large number of novel STs identified. Traditional phenotypic serotyping was required to type these isolates and the MLST database was modified and up-dated to incorporate the new data. Clearly, as we move forward the PHE MLST database will be constantly evolving and this data will be shared with colleagues in the field via existing MLST databases and their WGS compliant successors e.g., EnteroBase & BIGSdb. While it is difficult to draw conclusions based on our small sample size, MLST may not currently be an appropriate tool for the classification of *Salmonella* sub-species II–IV, due to the lack of a discrete population structure of EBGs. However, non-subspecies I isolates which are mainly adapted to cold blooded animals and/or reptiles contributed to less than 1.7% of the workload during the time frame of the study. Although MLST approach is generally more discriminatory than serotyping, it does not always provide the fine resolution required for public health surveillance. Further analysis based on single nucleotide polymorphisms in the core genome compared to a type strain representing the most common eBGs is performed for outbreak detection and investigation (*Ashton et al., 2015*).

In conclusion, serotyping inferred from 7-gene MLST results derived from WGS data is an accurate, robust, reliable, high throughput typing method that is well suited to routine public health surveillance of *Salmonella*. This approach supports the maintenance of traditional serovar nomenclature and provides further insight on the true evolutionary

relationship between isolates, as well as a framework for fine level typing within eBGs for surveillance, outbreak detection and source attribution.

## ACKNOWLEDGEMENTS

We would like to thank all the members of the *Salmonella Whole Genome Sequencing Implementation Group* including Steve Connell, Anna Lewis, Andy Levy, Clare Maguire, Clare Wend-Hansen, Martin Day, James Rogers, Siham Ibrahim, Arlene Barcenilla, Vineet Patel, Kiran Jayan, Anthony Underwood, Catherine Arnold and Ian Harrison.

### Funding

The research was partially funded by the National Institute for Health Research Health Protection Research Unit (NIHR HPRU) in Gastrointestinal Infections at the University of Liverpool in partnership with Public Health England (PHE), University of East Anglia, University of Oxford and the Institute of Food Research. The funders had no role in study design, data collection and analysis, decision to publish, or preparation of the manuscript.

### Grant Disclosures

The following grant information was disclosed by the authors:
National Institute for Health Research Health Protection Research Unit (NIHR HPRU).
Public Health England (PHE).
University of East Anglia.
University of Oxford.
Institute of Food Research.

### Competing Interests

The authors declare there are no competing interests

### Author Contributions

- Philip M. Ashton and Satheesh Nair conceived and designed the experiments, performed the experiments, analyzed the data, contributed reagents/materials/analysis tools, wrote the paper, prepared figures and/or tables, reviewed drafts of the paper.
- Tansy M. Peters conceived and designed the experiments, performed the experiments, reviewed drafts of the paper.
- Janet A. Bale conceived and designed the experiments, performed the experiments, analyzed the data, contributed reagents/materials/analysis tools, reviewed drafts of the paper.
- David G. Powell analyzed the data, contributed reagents/materials/analysis tools, reviewed drafts of the paper.
- Anaïs Painset analyzed the data, contributed reagents/materials/analysis tools, prepared figures and/or tables, reviewed drafts of the paper.
- Rediat Tewolde conceived and designed the experiments, performed the experiments, contributed reagents/materials/analysis tools, reviewed drafts of the paper.

- Ulf Schaefer contributed reagents/materials/analysis tools, reviewed drafts of the paper.
- Claire Jenkins analyzed the data, wrote the paper, prepared figures and/or tables, reviewed drafts of the paper.
- Timothy J. Dallman conceived and designed the experiments, analyzed the data, contributed reagents/materials/analysis tools, wrote the paper, reviewed drafts of the paper.
- Elizabeth M. de Pinna and Kathie A. Grant conceived and designed the experiments, reviewed drafts of the paper.

## DNA Deposition

The following information was supplied regarding the deposition of DNA sequences:

All data from the Salmonella surveillance project are deposited in the BioProject of the SRA: PRJNA248792. Per sample accessions are available in Table S1.

## Data Availability

MOST code for MLST and KmerID code are available here: https://github.com/phe-bioinformatics.

## Supplemental Information

Supplemental information for this article can be found online at http://dx.doi.org/10.7717/peerj.1752#supplemental-information.

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
