# Peer review of "Identification of Salmonella for public health surveillance using whole genome sequencing"

_PeerJ, doi:10.7717/peerj.1752_

## Round 0.1 · original submission · Minor Revisions

The manuscript presents sound biological findings in a clear, concise and well written form. Unfortunately, I agree with the opinion of reviewers 1 and 3 that further details in the Methods section are fundamental prior to final acceptance for publication. Particularly, I would encourage you to provide an extended description of the components of the bioinformatics pipeline. The text should be detailed enough to allow the implementation of the pipeline elsewhere and to ensure reproducibility of the results.

Furthermore, provide full references in the Methods section. Some of the methods are listed by what seems to be their PubMed id's.

Reviewer 1 ·

Basic reporting

Maybe trivial, but I do not understand why for some articles the author provides the pubmed id (.eg lines 131 and 137), please provide a the citation.

Experimental design

The bioinformatic analysis is extremely important for the results described in the manuscript.
It is important that the author explains the following points:
1) Explain the k-mer identification algorithm (if you wrote it yourself) or at least cite the method you used.
2) The author should describe the reasons that made him choose the bioinformatic tool described in Inouye et al. 2014 (SRST2) instead of the assembly based methods like BIGSdb (Jolley et al. 2010) or an approach similar to the one described in Larsen et al. 2012. Reading the manuscript about SRST2 Salmonella typing is done on only 42 reads sets, which is a very small dataset to evaluate the typing of Salmonella (while you have very nice set of data), please also refer to this article about Salmonella typing using WGS (Jolley et al. 2012, pubmed:22282518).
3) Please clarify the difference between the original SRST2 and the modified version you used.
4) Please, test (at least the small set with mismatches n=271) the typing using an assembly based method (you choose it...) and show the differences (if any) whith the one you used. I believe the test should not take more than 5 days on 8-16cpu workstation.
This pretty much goes back to point 1), and also give some information the computational time and the hardware you used.

Making clear how simple (or difficult) is to setup the bioinformatic workflow you used might influence the decision of other microbiologist/clinicians who are considering the use of WGS for surveillance. This might help spreading the use of WGS in public health institutions.

Validity of the findings

The findings might be of great contribution to scientific community, especially once the new MLST schemes are shared with other researchers.

Additional comments

As I just said in the box above, your findings might be of great contribution to scientific community. However, a more clear explanation of the bioinformatic workflow is required to help other researchers to embed the use of WGS in the workflow they use everyday.

Looking forward to receive the revised version of your manuscript.

best regards

Reviewer 2 ·

Basic reporting

No comments

Experimental design

No comments

Validity of the findings

No comments

Additional comments

The manuscript is clear, concise and well written. It is an important publication when working with public health surveillance of Salmonella in the midst of this transition period from traditional methods to whole genome sequencing. The manuscript is not adding surprising new findings to the insight of phylogeny of Salmonella.

Reviewer 3 ·

Basic reporting

"No comments"

Experimental design

1. Authors presented some phylogenetic trees but it was not mentioned how those trees had been constructed. This should be mentioned in method section.
2. K-mer identification step could use more explanation on the K-mer size and scores.

Validity of the findings

1. I would suggest to add a summary table with number of samples per subspecies, number of STs identified/failed to identify.

2. The aim of the article meets the conclusion.

In conclusion, authors mentioned that WGS MLST is the approach for fine level typing within outbreak detection.
Normally, Salmonella isolates from outbreak have the same ST type. If closely related Salmonella isolates from different outbreaks have the same ST type, can WGS MLST differentiate them?

Additional comments

In general, article is well written, authors tested high number of isolates.

---

## Round 0.2 · accepted · Accept

The current version of the manuscript addresses the comments / suggestions raised during the previous version satisfactorily.

Please, proofread your manuscript as some minor corrections might still be needed.

Reviewer 2 ·

Basic reporting

No comments

Experimental design

No comments

Validity of the findings

No comments